# Vulnerability of smallholder farmers to climate variability and change across different agro-ecological Zones in Oromo Nationality Administration (ONA), North east Ethiopia

**Ahmed Aliy Ebrahim** [1]☉*, **Birhan Asmame Miheretu**[1]☉, **Arragaw Alemayehu**[2]

**1** Department of Geography and Environmental Studies, Wollo University, Dessie, Ethiopia, **2** Department of Geography and Environmental Studies, Debre Berhan University, Debre Berhan, Ethiopia

☉ These authors contributed equally to this work.
* abufawuzan36@gmail.com

## Abstract

Ethiopia is frequently identified as a country that is highly vulnerable to climate variability and change. The study was aimed to examine agro-ecological based smallholder farmers' livelihood vulnerability to climate variability and change in Oromo Nationality Administration (ONA), North East Ethiopia. Data were collected from a survey of 335 sampled households, focus group discussion, and interview from three different agro-ecologies in the study area and secondary sources. Count, percentage, mean, standard deviation, Chi-square test (test of independence), ANOVA, Livelihood Vulnerability Index (LVI) and LVI-IPCC were used for analysis. LVI and LVI-IPCC results revealed that Kolla is the most vulnerable (0.18) because of its highest exposure (0.74) and sensitivity (0.71) values and lowest adaptive capacity (0.49) while Daga is least vulnerable (0.08) because of its lowest exposure (0.61) and sensitivity (0.42). Overall, results suggest that the two methods resulted in similar degrees of vulnerability and identified Kolla agro-ecological zone as the most vulnerable while the Dega agro-ecological zone is the least vulnerable of the three agro-ecological zones. The researchers conclude that development strategies and plans should be prepared considering local-specific issues and/or situation.

## Introduction

In recent decades, climate change have caused impacts on natural and human systems on all continents and across the ocean. In many regions, changing precipitation or melting snow and ice are altering hydrological systems, affecting water resources in terms of quantity and quality, terrestrial, freshwater, and marine species have shifted their geographic ranges, seasonal activities, migration patterns, abundances, and species interactions, and negative impacts on crop yields are some of the global impacts of climate change [1].

Africa is highly vulnerable to climate change on account of its large rural population that remains highly dependent on rain-fed agriculture for food, its natural resource-based

**Data Availability Statement:** The minimal data was uploaded in the supporting information section.

**Funding:** Amhara regional state funded this study. The funder had no role in study design, data collection and analysis, decisions to publish, interpretation of the data and preparation of the manuscript for publication.

**Competing interests:** The authors have declared that no competing interests exist.

economy, and constraints on internal trade [2]. According [1], climate change is a multiplier of existing health vulnerabilities including insufficient access to safe water and improved sanitation, food insecurity, and limited access to health care, education, economic insecurity, all being of particular concern for Africa so does Ethiopia.

Ethiopia is heavily dependent on rainfed agriculture. Its geographical location and topography in combination with low adaptive capacity entail a high vulnerability to adverse impacts of climate change [3–5] stated that food insecurity arising from occurrences of droughts and floods, the outbreak of diseases such as malaria, dengue fever, water-borne diseases (such as cholera, dysentery) associated with floods and respiratory diseases associated with droughts, and degradation due to heavy rainfall, damage to communication, road and other infrastructure by floods are the major climate variability related adverse impact affecting the nation.

Ethiopian smallholder farmers are facing different types of climate change-related risks, such as reduced or variable rainfall, warming temperatures, shortage of water, and flooding [3,4]. Oromo Nationality Administration (ONA) belongs to one of the most vulnerable areas to climate change and variability in the country [3–5]. Smallholder farmers in Oromo Nationality Administration are engaged in a subsistence production system where farms are already under significant climate stress [6]. Most households are already food insecure [3–5]. Drought contributes to reduced agricultural productivity, and the future sustainability of the sector depends on the types of adaptation strategies [4]. As a result, the present study considers smallholder farmers of the area.

Currently, livelihood vulnerability assessment has become a key focus of the scientific and policy-making communities. Previous studies [7–10] assessed small farmers' livelihood vulnerability assessment to climate variability and change based on agro-ecologies. However, few studies used an expert judgment approach in weighting indicators of vulnerability components. Hence, the methodology applies to other parts of Ethiopia as well.

As a result, this paper aimed to examine agro-ecological based smallholder farmers' livelihood vulnerability to climate variability and change in the Oromo Nationality Administration (ONA), North East of Ethiopia. The study expected to support policymakers, local practitioners, and farmers in the study area by providing local challenges of the study area and would help in adopting local specific climate adaptation strategies. It might have also a little contribution to the existing knowledge.

The study area is located in the arid and fragile landscape of Northeast Ethiopia where adverse effects of climate change and variability have contributed to the deterioration in the livelihood of farmers. The innovative element of this study is the assessment of vulnerability across agro-ecological zones and household data taken from Kebeles, which are the smallest units of administration in the country, which is useful for planning and implementation of climate change adaptation interventions at the grassroots level. Moreover, this household-level assessment of vulnerability is useful to identify and prioritize vulnerable areas and contributing factors of vulnerability.

## Material and methods

### Description of the study area

Geographically, the study area is found at 10˚5' to 11˚26'N and 39˚48'E to 40˚ 25' E (Fig 1). It has an area of 392, 684 hectares of land [11]. Based on [12], the study area falls into three climatic zones: hot (*Kolla*) 841 m.a.s.l (meter above sea level) to 1500m.a.s.l, temperate (*Woinadega*)1500m.a.s.l to 2300m.a.s.l, and cool (*Dega*) 2,300m.a.s.l to 2,838m.a.s.l. In terms of area, *Kolla* shares 2,609.472km$^2$ (63 percent share). The second share is that of *Woyenadega* which accounts 1,473.63km$^2$ (35 percent share). Besides, the smallest agro-ecological zone of the

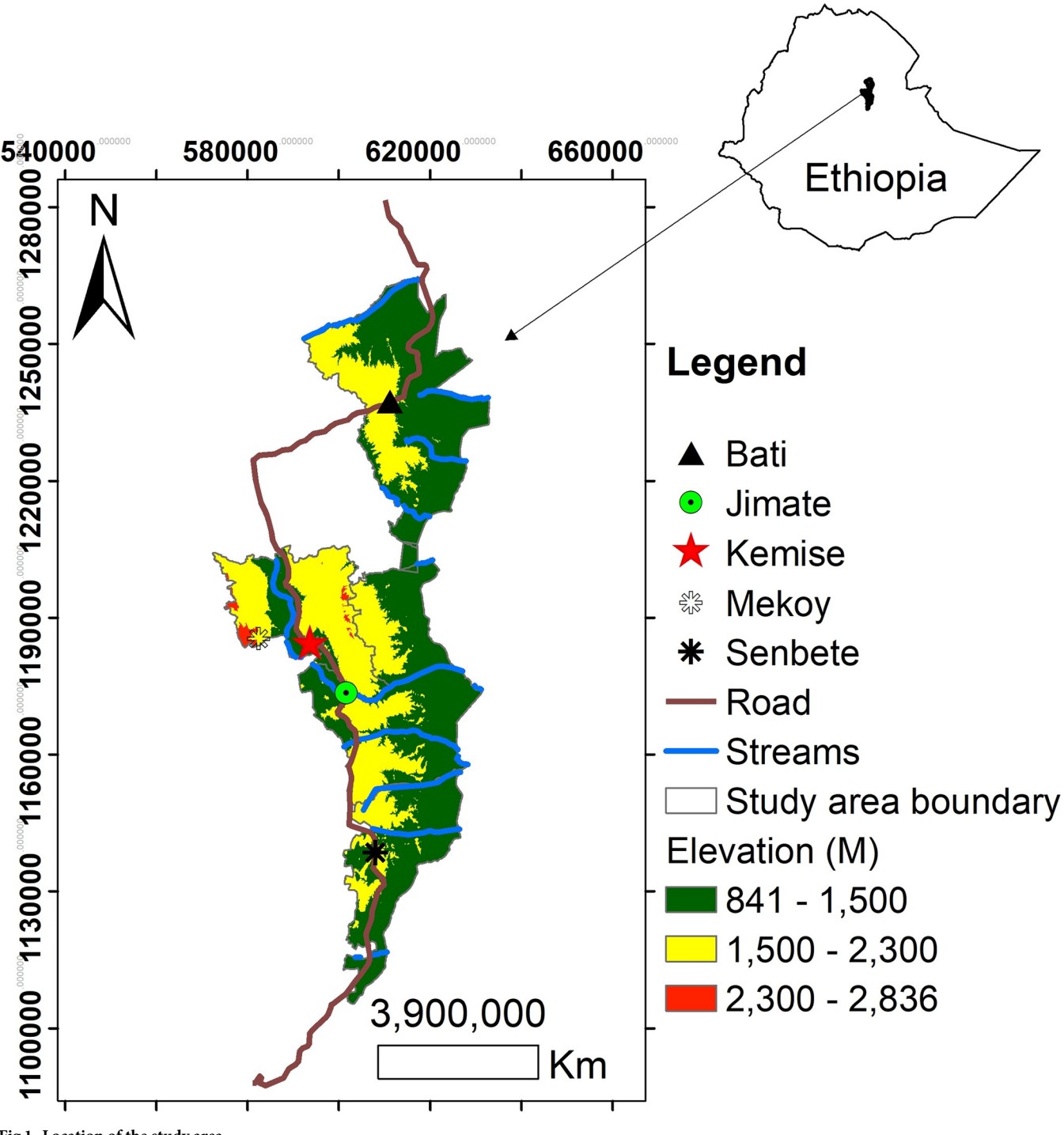

**Fig 1. Location of the study area.**

study area is *Daga* which has an area of 65 km$^2$ and only two percent of total share. Moreover, annual rainfall of the study area ranges from 610mm to 1,115mm with 856mm means [Own computation using NMA data, 2018/19]. The minimum, maximum, and mean temperature of the study area ranges from 16 to 19˚C, 27.7 to 29.8˚C, and 2.7 to 24.55˚C respectively [Own

calculation using NMA data, 2018/19]. Besides, according to [13] six major soil associations cover the landscape of the Oromo Nationality Administration. These are Lithosols, Eutric Cambisols (Lithic), Eutric Cambisols (Stony), Eutric Regosols (Lithic), Chromic Vertisols, and Orthic Solonchaks.

### Source: Department of Geography and Environmental Studies, Wollo University, Ethiopia

Oromo Nationality Administration is bounded by the South Wollo Zone of Amhara National Regional State (ANRS) both in north and west; by the North Shawa Zone of ANRS in south and west, and by Afar National Regional State in south and east. The study area is organized by two town administrations namely: Bati town, and Kemissie town administration, and five rural districts (*Woredas*) known as Dawachefa Woreda, Dawe Harewa, Artuma Fursi, Bati, and Jille Tumuga Woreda. _ Kemissie is main town, which is located 325kms north of Addis Ababa. The study area has 105 rural and 22 urban Kebeles (the smallest unit of administration). The total population of the study is estimated at 574,240 [14]. According to [15], 60,513 hectares is being ploughed annually within the study area. Sorghum, Maize, and Teff are major cereal crops that dominate the subsistence farming of the study area. Furthermore, livestock is another very crucial asset and livelihood of the rural people. Hence, 512, 715 equines, 261,529 goats and sheep, 38, 125 cattle, and 14,575 camels are found in the study area [15]. In addition, 16 percent of urban people in the ONA are based on non-farm and off- farm activities such as small enterprises, services, trade, manufacturing, and agro processing projects as well as investment areas [16].

### Research design

**Data set.** A mixed approach with a cross-sectional survey was undertaken. The rationale for using both quantitative and qualitative data was to obtain statistical results from a sample and follow up with a few individuals to explore those results in-depth as well as to describe and report easily. Qualitative and quantitative data types were collected using primary and secondary data sources. Primary data sources were household survey which was given priority, key informant interview, and focus group discussion. Secondary data sources include journals, reports, and plans, as well as internet websites. Hence, participants of key informant interviews were 12 in number and selected purposely after the consultation of developmental workers of each *Kebele*. On the other hand, FGD was undergone at *Kebele* and *Woreda* level with nine FGD groups with 90 participants. This includes six FGDs at each Kebele (ten participants in each group): two Kebeles in Kolla, three Kebeles in woyina Daga, and one Kebele from Daga agro-ecology zones, and three FGD groups at selected Woredas proportion to each agro-ecology zone (ten members in each group).

Multi-stage stratified sampling was employed in the study. First, using 30 m resolution raster DEM the seven woredas of ONA were grouped into 3 strata based on their agro-ecological characteristics. Then, one woreda was selected from *Kolla* and *Woyinadaga* agro-ecological zones using a simple random sampling method. From *Daga* agro-ecology, Dawachafa *Woreda* was selected purposely as it is the only woreda that has *Daga* agro-ecology. In the second stage, six Kebeles (the smallest unit of administration) were selected from each selected *Woreda*: one *Kebele* from *Dega* (highland), two *Kebeles* from *Kolla*, and three *Kebeles* from *Woyinadega* agro-ecology were randomly selected in proportion to the number of kebeles and households. Eventually, households were selected using a systematic random sampling technique proportionate to the size of households living in each Kebele. The sampling frame was list of households that were obtained from the Kebele Administration Office (KAO).

The sample size was determined using the formula constructed by [17]

$$n = (Z^2 * p * q * N)/(N - 1) + Z^2 * p * q \tag{1}$$

Where, n is the desired ample size for the study; z is the upper points α/2 of standard normal distribution at 95% confidence level which is equal to 1.96; e is acceptable error at a given precision rate (assumed 5%); p is the proportion of households (which is taken as 0.5 or 50%) —most conservative case that 'n' will be the maximum and the sample will yield at least the desired precision; q is 1-p; N is the total households in the sampling frame equals to 6,347.

$$n = \frac{(1.96)^2 + 0.5 * 0.5 * 6347}{(0.5)^2(6347 - 1) + (1.96)^2 * 0.5 * 0.5} = 362.4 \sim 363$$

As a result, the sample size of this study was 363 households. However, because of the refusal of the respondents, missing data, and low response rating the following were missed: one household (HH) from *Kolla*, two HH from *Woyinadega*, and 25 households from *Daga* 28 households. Hence, 335 sample households were analyzed in the study.

## Vulnerability analysis

This study employed the Livelihood Vulnerability Index (LVI), and LVI-IPCC indicator approach as applied in [18]. The vulnerability assessment approach has prevalence advantages. Firstly, it is a practical method in monitoring trends and exploring conceptual framework. Second, it incorporates both socioeconomic and environmental factors. Third, easier to compute and comprehend by readers with low mathematical inclination. Fourth, provides the projection of future vulnerability for effective adaptation planning. Fifth, provides a detailed depiction of factors driving a household's livelihood vulnerability in a particular study area. Sixth, provides a framework for analyzing the key components that makeup livelihoods, a practical method to identify vulnerable communities, to prioritize the potential interventions recommended to policymakers, local authorities, and development organizations [10,19,20] However, the basic challenge in constructing indices is lack of standard ways of assigning weight to each other [21].

According to [22], there are three methods for assigning weight to indicators: using expert judgment, applying the arbitrary choice of equal weight, and using a statistical method such as Principal Component Analysis (PCA) or Factor Analysis (FA). In this study, expert judgment was applied since it has been little used by others despite subjectivity being a challenge yet. The use of different techniques of vulnerability assessment simultaneously can add value to methodological contribution. Of course, each technique has its advantage and disadvantages. After critically reviewing available research, the present study preferred to use expert judgment as it gives more freedom to know who is vulnerable to what. As the extent of vulnerability and options for adaptation vary across space and in time, the use of expert judgment helps to understand local scale vulnerability context [5].

Since the sub-components are measured on a different scale, they were first standardized using Eq (2).

$$IndexSr = \frac{Sr - Smin}{Smax - Smin} \tag{2}$$

Where, Sr is the observed sub-component indicator for agro-ecology r and Smin and Smax are the minimum and maximum values, respectively. But, when a higher value is good and has a positive contribution to minimizing vulnerability (like educational status, access to microfinance and etc), the normalized value for each indicator can be computed by (1-index Si) or the

formula should be inverted using Eq [3]:

$$IndexSr = Smin - Sr/Smax - Smin \qquad (3)$$

After standardizing each indicator, the value of each dimension was calculated using Eq (4):

$$Mr = \sum\nolimits_{i \neq 1}^{n} index\ Sr * W / \sum W \qquad (4)$$

Where, Mr is one of the 13 major components (Socio demographic, infrastructure, housing, technology, social networks, livelihood strategies, asset, food, health, natural disaster, climate variability, water, agriculture, natural resources, soil and water) for agro-ecology r; indexsri, represents the sub-components indexed by i, that make up each major component, and W is averaged weight of each component which had been given by 14 experts.

Once values for each of the major components were calculated, they had averaged to obtain the total level of LVI using Eq (5):

$$LVIr = \sum\nolimits_{i \neq 1}^{n} WmiMri / \sum\nolimits_{i \neq 1}^{13} WMi \qquad (5)$$

Where, LVI is the Livelihood Vulnerability Index for agro-ecology r, equals the weighted average of the 13 major components. The weights of each major component, $W_{Mi}$, were determined by the number of sub-components that make up each major component and were included to ensure that sub-components contribute as the weight given to the overall LVI. LVI scaled from 0 (least vulnerable) to 1 (most vulnerable).

Furthermore, according to [18], LVI-IPCC index is calculated by taking the three important parameters of vulnerability used by IPCC namely exposure, sensitivity, and adaptive capacity. Thus, it was computed using the following Eq (5).

$$CFR = \sum\nolimits_{i \neq 1}^{n} WmiMri / \sum\nolimits_{i \neq 1}^{n} WMi \qquad (6)$$

Where, CFr, is an IPCC-defined contributing factor (exposure, sensitivity, or adaptation capacity) for agro-ecology r, $M_{ri}$ M is the major component for agro-ecology r indexed by i, $W_{Mi}$ is the weight of each major component, and n is the number of major components in each contributing factor.

Finally, once exposure, sensitivity, and adaptation capacity were being calculated, the three contributing factors were combined using Eq (6):

$$LVI - IPCCr = (E = er - ar) * sr \qquad (7)$$

Where, LVI-IPCC is the LVI for agro-ecology r expressed using the IPCC vulnerability framework, $e_r$ is the calculated exposure score for agro-ecology r (equivalent to the natural disaster and climate variability major component), $a_r$ is the calculated adaptation capacity score for agro-ecology r (weighted average of the socio-demographic, infrastructure, housing, technology, social networks, livelihood strategies, asset, food, and health) and $s_r$ is the calculated sensitivity score for agro-ecology r (weighted average of water, agriculture, natural resources, soil and water)

The LVI-IPCC value was scaled from -1 to +1: where: -1 denotes least vulnerable (adaptive capacity is more than exposure and sensitivity), 0 denotes moderately vulnerable (exposure and adaptive capacity are equal) and 1 denotes extremely vulnerable (exposure and sensitivity are very high than adaptive capacity).

## Indicators of vulnerability

Since the model variables are constructed based on the IPCC definition of vulnerability, which are a function of adaptive capacity, sensitivity, and exposure, then the indicators to construct vulnerability function, is categorized under these three factors as follows.

## Adaptive capacity indicators

Adaptive capacity is defined as the ability of people and institutions to anticipate, withstand, and respond to climate change and variability, and to minimize, cope with, and recover from climate-related impacts. It refers to the whole of capabilities, resources (biophysical and socio-economic), and institutions of a country or region or individuals to implement effective adaptation measures [1]. It is all the ability to build resilience and reducing vulnerability. Consequently, in the study adaptive capacity has been determined by the following major components as indicated in Table 1.

**Table 1. Adaptive capacity major components, sub-components, and their hypothesized effect with vulnerability.**

| Major components | Subcomponents | Hypothesized functional relationship between indicator and vulnerability |
|---|---|---|
| Socio demographic | Male of the HH in % | - |
| | Age of the HH in years | + |
| | Capable of reading and writing (%) | - |
| | % of HHs who did not live with orphans | - |
| | HHs who received training to cope with climate change in % | - |
| Institutional capacity | Distance to market in km | + |
| | Distance to all weather road in km | + |
| | Distance to nearest school km | + |
| | Access to credit in % | - |
| | Access to radio in% | - |
| | Access to mobile phone in % | - |
| | HHs with housing not easily affected by climate related disasters in % | - |
| Technology | Insecticide and pesticide users in % | - |
| | Users of artificial fertilizer users in % | - |
| | Improved seed users in % | - |
| Social networks | IDIR membership in % | - |
| | FCA (%) | - |
| | HHs that did not go to government for any kind of assistance in the last past 12 months in % | + |
| Livelihood strategy | Agricultural gross income | + |
| | Non/off farm gross income | - |
| Asset | Livestock ownership (Z-score) | - |
| | Farm land size in hectare | - |
| | Parcent of HHs with fertile soil | - |
| Food | Irrigated land hectare | - |
| | % HH with enough food throughout the year | - |
| | % HH who produce their own food | - |
| Health | % HH who did not Chronic ally ill | - |
| | Distance to health center in kilometer | + |
| | HH who did not miss work or school in the past six months due to illness in % | - |

**Human capital.** The skills, knowledge, nutrition, ability to labour, and good health are important to the ability to pursue different livelihood strategies. Thus, in the study the following were considered: Lower access /availability of food, higher dependent on-farm food, lower access to health, education, and training, frequent or chronic and frequent communicable illness, and lass diversified livelihood strategies lead to low adaptive capacity or vice versa.

**Financial capital.** The financial resources which are available to people (savings, supplies of credit, remittances or pensions, insurance, social security) and which provide them with different livelihood options. Hence, the assumption here is access to saving and credit, low livestock ownership and crop saving, non-agricultural and/or agricultural income sources, HHs with no jobs during extreme events happen bring lower adaptive capacity or vice versa.

**Physical capital.** The basic built-up infrastructures (transport, water, irrigation, energy, houses, and communication) and the production tools and equipment which enable people to pursue their livelihoods. Therefore, included variables in the study are: insecticide and pesticide supply, fertilizer supply, improved seed supply, irrigation potential, access to market, all-weather road, school, veterinary services, radio/telephone, production means. As a result, the lower asset of livestock, irrigation potential, and access to technology leads to lower adaptive capacity.

**Social capital.** The networks and connections (shared values and norms, common rules and sanctions, collective actions, patronage, kinship, membership of groups, relationships of trust, mutual understanding and support, access to wider institutions of the society) upon which people draw in pursuit of livelihoods. The following have been considered in the paper: IDIR membership, HHs that have not gone to local government for any kind of assistance in the past 12 months, IQUB membership, and government assistance. It has been hypothesized that lower social networks result in lower adaptive capacity or vice versa.

**Natural capital.** The natural resource stocks from which resources flow useful for livelihoods are derived (example, land, water, grazing land, and environmental resources). Landless households and low farm size may have the potential to lower adaptive capacity or vice versa.

**Socio-demographic features.** Gender of the Household (HH) head, age of the HH head, households (HH) with orphans, and male/female-headed HHs, and education and training were considered under this major component. HHs with orphans, being old, female headed HHs, low education, and training lead to lower adaptive capacity or vice versa.

## Exposure indicators

Exposure is the presence of people, livelihoods, species or ecosystems, environmental functions, services, resources, infrastructure, or economic, social, or cultural assets in places and settings that could be adversely affected. Two sub-components that were considered to measure the exposure of smallholder farmers on their livelihood based on their agro-ecology are: natural disaster and climate variability. As Table 2 shows, natural disasters (extreme weather

**Table 2. Exposure major components, sub-components and their hypothesized effect with vulnerability.**

| Major components | Subcomponents | Hypothesized functional relationship between indicator and vulnerability |
|---|---|---|
| Natural disasters | Frequency of flood and drought in the last 30 years (2007–2016) | + |
| | Percent of HH who experienced crop failure | + |
| | Percent of HH who did not receive warning | + |
| | Injury and death during extreme events (in %) | + |
| | Households (HH) with no jobs during extreme events in% | + |
| Perception of Climate change | Percent of HH agreed on temperature is increasing | + |
| | Percent of HH agreed on rainfall is decreasing | + |

events) were measured using floods and drought over the last 10 years, warning about natural disasters, injury or death, and landslide events while climate variability was considered by the perception of smallholder farmers on change in temperature and rainfall in 10 years. Thus, It has been hypothesized that larger change/variability or frequency of floods, droughts, landslides, temperature, and/or rainfall, as well as low warning result in higher exposure.

### Sensitivity indicators

Sensitivity is the degree to which a system or species is affected either adversely or beneficially by climate variability or change. The effect may be direct (e.g., a change in crop yield in response to a change in the mean, range, or variability of temperature) or indirect (e.g., damages caused by an increase in the frequency of coastal flooding due to sea level rise [1].

To measure the sensitivity level of agro-ecological based smallholder farmers' livelihood vulnerability to climate variability and change, considered sub-components are: access/availability of water, consistent water supply, change in annual crop production and productivity, crop diversity, save seeds, farm management training, HHs depending on or exploiting natural resources, HHs using only forest-based energy for cooking, HHs reporting that firewood is being scarce now in comparison to the past 10 years, and Soil and Water Conservation (SWC). It has been assumed that lower access to water, dependency on natural resources, decrease the change of crop production, degradation of land, forest-based energy result in higher sensitivity (Table 3).

### Ethics approval and consent to participate

Ethical clearance was obtained from the ethical review committee of Wollo University College of Medicine and Health Sciences. Permission to conduct the study was obtained from the Oromo Nationality Administration (ONA) Zone Agriculture office. Before the data collection, the purpose of the study was explained to the study participants, and assurance was given that their participation in the study was voluntary. Then, informed written consent was obtained from each study participant. The confidentiality of the study participants' responses was ensured by not disclosing any information to a third party.

## Results and discussions

### Rainfall and temperature variability and trends

Annual rainfall exhibits a statistically significant decreasing trend in the period 1987–2016. March-May (Belg) and October–February (Baga) rainfall show statistically significant

**Table 3. Sensitivity major components, sub-components, and their hypothesized effect with vulnerability.**

| Major components | Subcomponents | Hypothesized functional relationship between indicator and vulnerability |
|---|---|:---:|
| **Water** | HHs not access to clean water in parcent (%) | + |
| | % of HHs with not consistent water supply | + |
| **Agriculture** | Decreasing of total production (quintals) | + |
| | % of HH perceived decreasing of crop yield | + |
| | % HH who did not apply diversification | + |
| | % HH who did not save seed | + |
| | % HH who did not have training on farm management | + |
| **Natural resources** | Percent of HH exploit natural resources | + |
| | Percent of HH using only forest based energy for cooking | + |
| | HH reporting that firewood is decreasing in the last 30 years | + |
| | Percent of HH reporting land degradation by climate change | + |

decreasing trends. While June-September (Kiremt) rainfall exhibits a statistically significant increasing trend. Rainfall shows moderate concentration in the area. Both the maximum and minimum temperatures show statistically significant increasing trends. However, significant spatiotemporal variability in the maximum and minimum temperatures is observed across the study stations at p = 0.05 level.

## Farmers' perceptions of climate change and variability

Farmers were asked whether or not they noticed changes in rainfall and temperature in their surroundings. Accordingly, most farmers perceived declining trends in rainfall and warming trends in temperature. Almost all farmers perceived that they have lost Belg and Bega rains, which are important for crop and livestock production. Farmers' perception of declining trends in annual and seasonal rainfall is consistent with results from meteorological data except for Kiremt rainfall which exhibits a statistically significant increasing trend. In terms of temperature, a warming trend was perceived by most farmers, and this was supported by meteorological data.

## LVI results

**Adaptive capacity: Socio-demographic, institutional capacity, technology, social networks, livelihood strategy, asset, food, and health.** *Socio-demography*. Socio-demography is compiled from five subcomponents: Male household (HH) headed, age of the household in years, education status (capable of reading and writing), percent of households who did not live with orphans, and HHs received climate change training (Table 4). In terms of socio-demography Kolla is most vulnerable (0.47) followed by Daga (0.39), and Woyina Daga (0.38). Kolla is vulnerable in socio-demography because of a low percentage of educational status and training on climate change. This finding agreed with the previous studies [8,10] who noted that component has a higher vulnerability effect in Kolla than Dega and Woyina Dega (0.4619). Inferential statistics revealed a significant difference among the three agro-ecologies in educational status at p < 0.001 but an insignificant difference in their age (Table 7).

*Institutional capacity (Infrastructure)*. Regarding institutional capacity, it was organized from seven indicators: distance to market, all weather-road, and school in kilometers, access to credit, radio, and housing. *Kolla* zone is most vulnerable followed by *Woyinadaga* and *Daga* which scored 0.52, 0.39, and 0.31 respectively (Table 4). Kolla agro-ecology became more vulnerable due to distance to market and all-weather roads (in km), access to radio and mobile phone which scored 0.84, 0.75, 1.71, and 0.84 respectively. Similarly, [8] and [10] stated that Kolla households have a higher vulnerability score than Woyina Dega and Dega households. In addition, the researchers reject the null hypotheses (Ho: no difference in the agro-ecology zones) in terms of distance from home to all-weather roads (in km), distance from home to health services (in km), distance from home to market (in km), and distance from home to veterinary services (in km) at p < 0.001. The researchers accepted the null hypotheses (Ho: no statistically significant difference was found among agro-ecology zones) in distance from home to nearest school (in km), as well as the distance from home to the water source (in km) as p > 0.05 (Table 7).

*Technology*. The component was formed from three indicators: insecticide and pesticide users, users of artificial fertilizer, and improved seeds. In the sample households' users of artificial fertilizer were 55.3% in *Kolla*, 45% in *Woyinadaga*, 51.6% in *Daga* agro-ecology. Whereas users of insecticide and pesticide: 55.3% in *Kolla*, 97.5% in *Woyinadaga*, and 71.1% in *Daga*. Users of improved seeds were: 24.7%, 82%, and 60.2% in Kolla, Woyina Daga, and Daga respectively. In terms of technology, though it seemed better in using of artificial fertilizer

**Table 4. Major components' scores, indicator index, and weighted average of adaptive capacity profile.**

| Subcomponents | Factional relationship with vulnerability | Agro-ecology | | | Major components | Agro-ecology | | |
|---|---|---|---|---|---|---|---|---|
| | | Kolla | Woyina Daga | Daga | | Kolla | Woyina Daga | Daga |
| | | Index*weighted value | Index*weighted value | Index*weighted value | | | | |
| Male of the HH in % | - | 0.23 | 0.43 | 0.43 | Socio demographic | 0.47 | 0.38 | 0.39 |
| Age of the HH in years | + | 1.23 | 1.49 | 1.37 | | | | |
| Cable of reading and writing (%) | - | 2.20 | 1.95 | 1.28 | | | | |
| % of HHs who did not live with orphans | - | 0.48 | 0.50 | 0.62 | | | | |
| HHs who received training to cope with climate change in % | - | 1.96 | 0.58 | 1.40 | | | | |
| Distance to market in km | + | 0.84 | 0.72 | 0.22 | Institutional capacity | 0.52 | 0.39 | 0.31 |
| Distance to all weather road in km | + | 0.75 | 0.30 | 0.37 | | | | |
| Distance to nearest school km | + | 0.22 | 0.57 | 0.26 | | | | |
| Access to credit in % | - | 0.65 | 0.80 | 0.92 | | | | |
| Access to radio in% | - | 1.71 | 0.98 | 0.77 | | | | |
| Access to mobile phone in % | - | 0.84 | 0.56 | 0.60 | | | | |
| HHs with housing not easily affected by climate related disasters in % | - | 2.02 | 1.25 | 0.97 | | | | |
| Insecticide and pesticide users in % | - | 0.92 | 0.03 | 0.50 | Technology | 0.56 | 0.27 | 0.40 |
| Users of artificial fertilizer users in % | - | 0.97 | 1.21 | 1.06 | | | | |
| Improved seed users in % | - | 1.30 | 0.31 | 0.69 | | | | |
| | - | 0.75 | 0.69 | 0.60 | Social networks | 0.61 | 0.53 | 0.63 |
| FCA (%) | - | 1.42 | 1.15 | 1.28 | | | | |
| HHs that did not go to government for any kind of assistance in the last past 12 months in % | + | 0.65 | 0.58 | 1.02 | | | | |
| Agricultural gross income | + | 0.64 | 0.62 | 0.48 | Livelihood strategy | 0.42 | 0.43 | 0.38 |
| Non/off farm gross income | - | 1.56 | 1.64 | 1.49 | | | | |
| Livestock ownership (Z-score) | - | 1.95 | 1.99 | 1.63 | Asset | 0.76 | 0.78 | 0.72 |
| Farm land size in hectare | - | 1.50 | 1.53 | 1.80 | | | | |
| % of HH with fertile soil | - | 1.67 | 1.64 | 1.58 | | | | |
| Irrigated land in hectare | - | 1.18 | 1.36 | 0.96 | | | | |
| % HH with enough food throughout the year | - | 1.69 | 1.41 | 1.34 | Food | 0.42 | 0.34 | 0.38 |
| % HH who produce their own food | - | 0.18 | 0.11 | 0.34 | | | | |
| % HH who did not Chronic ally ill | - | 0.60 | 0.41 | 0.33 | Health | 0.12 | 0.16 | 0.22 |
| Distance to health center in km | + | 0.53 | 0.67 | 1.38 | | | | |
| HH who did not miss work or school in the past six months due to illness in % | - | 0.10 | 0.63 | 0.30 | | | | |
| HH who did not suffer from communicable diseases in % | - | 0.12 | 0.03 | 0.37 | | | | |

Index*weighted value = = since indicators were given weight by expert judgment each index value of an indicator was multiplied by each weighted value of the same indicator that scaled by experts.

(55.3%), Kolla scored 0.56 (more vulnerable), Woyina Daga 0.27 (less vulnerable), and Daga is moderate scoring 0.40 respectively (Table 4). This finding has aligned with [8,23] who revealed that the least technology usage entails lower adaptive capacity and higher vulnerability of lowland (Kolla) AEZ. The area is arid and semi-arid, with limited agricultural adaptation

technology usage. Options for adaptation are relatively low. This is also confirmed a previous study [4]. That is why only three technology indicator are used. However, we understad that use of many variable beer better results.

*Social network*. The social network is an amalgamation of IDIR membership, Farmers Cooperative Association (FCA), and government assistance. *Kolla*, *Woyinadaga*, as well as *Daga* had 0.61, 0.53 and 0.63 index values of social networks respectively (Table 4). What could be concluded is that *Daga* is most vulnerable in the social network. This finding disagreed with the studies [8,10] who found that households in the dry lowland agro-ecology are more vulnerable in terms of social network. *Woyinadga* is less vulnerable. *Dag*a is vulnerable in the study area because of a lack of assistance from the government. Even though Daga is better in Organization membership (68%, and 20% engagement in IDIR and FCA), it lacks government assistance with 10% share while Kolla and Woyina Daga have got government assistance 44% and 49% in a better way.

*Livelihood strategies*. Agricultural and non/off-farm annual gross incomes are the two sub-components of livelihood strategy. Fortunately, the three AEZs had scored 0.42, 0.43, and 0.38 in Kolla, Woyina Daga, and Daga (Table 2). Comparatively, *Woyinadaga* is more vulnerable in terms of livelihood strategy than the rest. Besides, Kruskal Wallis Test revealed a statistically significant difference in annual gross income (both farm and non/off-farm gross incomes) of sampled households across AEZ (Table 7). This result is in line with [24]. However, the result did not agree with [8] who indicated that with an index value of 0.8998 livelihood strategies have had a higher effect on Kolla vulnerability, than in Daga, and woyina Daga agro-ecology.

*Asset*. Wealth major component was compiled using livestock ownership (Z-score), farm and irrigated land size (in hector), and fertile soil of the farmland. The asset had a scale of 0.76 in Kolla 0.78 in Woyina Daga (more vulnerable), and 0.72 in *Daga* which is least vulnerable. Significance difference was found across AEZs in irrigated and farmland, as well as soil fertility status of farmland, and Z-score of animal resources (Table 4). Crop production and livestock constitutes a significant source of income for farmers in many parts of Ethiopia [4,5]. Livestock ownership, irrigated land size and soil fertility of the farm land are the major source asset in the area. However, the contribution of off farm income, remittance and trees are undeniable. That is why the present study gives priority on the important sources of asset.

*Food*. Two indicators were found under this major component. One is HHs with enough food throughout the year, and the second was HHs who yield their food. Despite 92% of residents of *Kolla* yield their own food, only 22% of them had enough throughout the year which is lowest comparing with 36% and 39% of *Woyinadaaga*, as well as *Daga* dwellers. Thus, *Kolla* (0.42) is more vulnerable while *Woyinadaga* is less vulnerable (0.34), and Daga is moderate (0.38) in terms of food major components (Table 2). The researchers computed a one-way ANOVA comparing the number of months respondents are food insufficient among the three agro ecology zones (F (2,225): 19.602, p < 0.05). Tukey HSD was used to determine the nature of the difference between AEZ. The analysis revealed that Kolla had more food insufficient months (m: 61, sd: 1.64) than *Woyinadaga*, and (m: 4.3, sd: 1.83) Daga (m: 5, sd: 1.67). Besides, *Woyinadaga* residuals had less number of food insufficient months (m: 4.3, sd: 1.83) than *Daga* (m: 5, sd: 1.67).

*Health*. This major component was computed using distance to reach to the nearby health center (in km), and percent of HHs who did not miss work or school in the past six months due to illness, HH who did not suffer from communicable diseases, and HHs who did not chronically ill. The health profile index value ranges from 0.12 (in Kolla) to 0.22 (Daga) (Table 4). Daga is most vulnerable in health profile index value. Statistically, significant difference was found across AEZs regarding distance to reach to the nearby health center (Table 7) at p < 0.01.

**Exposure: Natural disaster and perception of households on climate change (rainfall and temperature).** *Natural disaster.* This major component is a combination of frequency of flood and drought in the last 10 years, households (HHs) who experienced crop failure, who did not receive a warning, who faced injury and death during extreme events, and HHs with no jobs during extreme events. The index values of each AEZ were: *Kola* (0.50), *Woyinadaga* (0.43), and *Daga* (0.32) (Table 5). The result indicated *Kolla* is more vulnerable to natural disasters while *Dag* is less. Vulnerability of *Kolla* agro-ecology due to the frequency of flood and drought in the period 1987–2016, percent of HHs who experienced crop failure, who did not receive a warning, and injury and death during extreme events with 1.14, 2.46, 1.81, and 0.17, and 1 values respectively. Moreover, Kruskal Wallis Test showed a statistical significant difference across agro-ecology zones in Frequency of crop failure as well as drought/flood in the last 30 years (1987–2016) (Table 7).

*Climate variability/change.* Climate variability/change of exposure was computed using smallholder farmers' perceptions on rainfall and temperature change. As a result, it accounts in Kolla 0.98, in Woyina Daga 0.98, Daga 0.9 (Table 5). Hence, *Kolla* and *Woyinadaga* area is equally considered highly vulnerable in climate variability/change major component. Small-holder farmers' perception toward climate variables indicated that temperature is increasing and rainfall is decreasing though slight variation across agro-ecological zones was found. The finding revealed that *Kolla* agro-ecology is more vulnerable than *Daga* in terms of exposure profile. The same result was found by [8,23,24] who noted that *Kolla* agro-ecology is found to be more vulnerable.

**Sensitivity: Water, agriculture, and natural resources.** *Water.* Households (HHs) who have not access to clean water and HHs with inconsistent water supply are the elements constructed water major component. The water profile index was scaled 0.72, 0.32, and 0.41 in *Kolla*, *Woyinadaga*, and *Daga* agro-ecology zone respectively (Table 6). This showed *Kolla* agro-ecology is highly determined by water major component while it has less effect on *Woyinadaga* agro-ecology. *Kolla* was vulnerable because of inaccessibility and inconsistency of water supply with an index of 1.30 and 1.39 respectively. Therefore, due attention should be

**Table 5. Major components' scores, indicator index, and weighted average of Exposure profile.**

| Subcomponents | Factional relationship with vulnerability | Agro-ecology | Major components | Agro-ecology | Subcomponents | Factional relationship with vulnerability | Agro-ecology | Major components |
|---|---|---|---|---|---|---|---|---|
| **Frequency of flood and drought in the last 30 years (1987–2016)** | + | 1.14 | 1.00 | 0.80 | Natural disaster | 0.50 | 0.43 | 0.32 |
| **% of HH who experienced crop failure** | + | 2.46 | 2.08 | 0.82 | | | | |
| **% of HH who did not receive warning** | + | 1.81 | 1.37 | 1.73 | | | | |
| **Injury and death during extreme events in %** | + | 0.17 | 0.15 | 0.15 | | | | |
| **Households (HH) with no jobs during extreme events in %** | + | 1.22 | 1.22 | 0.85 | | | | |
| **% HH agreed on temperature is increasing** | + | 2.18 | 2.13 | 2.02 | Climate variability | 0.98 | 0.98 | 0.90 |
| **% of HH agreed on rainfall is decreasing** | + | 3.36 | 3.43 | 3.09 | | | | |

**Table 6. Major components' scores, indicator index and weighted average of Sensitivity profile.**

| Subcomponents | Factional relationship with vulnerability | Agro-ecology | Major components | Agroecology | Subcomponents | Factional relationship with vulnerability | Agro-ecology | Major components |
|---|---|---|---|---|---|---|---|---|
| HHs not access to clean water (%) | + | 1.30 | 0.23 | 0.37 | Water | 0.72 | 0.32 | 0.41 |
| % HH with not consistent water supply | + | 1.39 | 0.97 | 1.15 | | | | |
| Decreasing of total production (quintals) | + | 1.05 | 0.66 | 0.60 | Agriculture | 0.58 | 0.30 | 0.41 |
| % of HH perceived decreasing of crop yield | + | 2.22 | 2.15 | 1.63 | | | | |
| % HH who did not apply diversification | + | 0.96 | 0.28 | 0.70 | | | | |
| % HH who did not save seed | + | 1.70 | 0.18 | 0.56 | | | | |
| % HH who did not have training on farm management | + | 0.97 | 0.31 | 1.35 | | | | |
| %HH exploit natural resources | + | 1.15 | 0.69 | 0.48 | Natural resources | 0.82 | 0.75 | 0.44 |
| % of HH using only forest based energy for cooking | + | 2.64 | 2.27 | 0.80 | | | | |
| HH reporting that firewood is decreasing in the last 30 years | + | 2.74 | 2.80 | 1.48 | | | | |
| % of HH reporting land degradation by climate change | + | 2.72 | 2.77 | 2.21 | | | | |

given to solve water stress in *Kolla* areas of Oromo Nationality. In addition, a discussion at *Lugo Burka* and Garbi *Mudi Wachu* (*Kolla* sampled *Kebeles)* seriously discussed the scarcity of water, and they sincerely asked for government intervention. Our findings corroborated with a previous study conducted by [5].

*Agriculture.* Households (HHs) perceive decreasing of crop yield, decreasing of total production (in quintals), who did not apply diversification, HHs who did not save seed, and HHs who did not have training on-farm management are sub-components of agriculture major component. The major perceived factors contributed to the decrease in crop yield in the study area are the occurrence of droughts and floods, land degradation and declining soil fertility, socioeconomic, and demographic constraints. In Kolla, 98 percent of HHs perceived a decrease in crop yield, while in Woyinadaga and Daga, 95 percent and 72 percent of HHs perceived a decrease in crop yield, respectively.Moreover, 56% of HHs did not apply crop diversification in *Kolla* area whereas it was 13% and 33% in *Woyinadaga* and *Daga*. On the other hand, 31% of sampled households of *Kolla* had no training on-farm management while it was only 10% in *Woyinadaga*. About 67% of HHs did not save seeds in *Kolla* while 7% and 23% of sampled households saved seeds in *Woyinadaga* and *Daga*. Because of these variations among agro-ecology it could be said that *Kolla* agro-ecology was highly vulnerable by agriculture profile with an index value of 0.58 followed by *Daga* (0.41) and *Woyinadaga* (0.3 i.e. less vulnerable) (Table 6). Furthermore, inferential statistics indicated that statistically significant difference among AEZs in terms of decreasing of total production (in quintals) $p < 0.001$ (Table 7). This finding had an agreement with [8] who concluded that agriculture component has the largest contribution to the vulnerability of the community in *Kolla* with a value of

**Table 7. Explanatory and ordinal (dependent) variables considered in the Kruskal Wallis Test analysis for the three agro-ecology zones (AEZ).**

| Variables | Significance level |
|---|---|
| Age of respondents in years | 0.68 |
| Annual gross income | 0.000 |
| Annual remittance income | 0.007 |
| Average amount of credit annually (Eth Birr) | 0.000 |
| Average amount of saving in a year (Eth Birr) | 0.178 |
| Decreasing of crop yield (quintal per hectare) | 0.000 |
| Distance from home to all- weather roads (in km) | 0.000 |
| Distance from home to health services (in km) | 0.000 |
| Distance from home to market (in km) | 0.005 |
| Distance from home to nearest school (in km) | 0.294 |
| Distance from home to veterinary services (in km) | 0.000 |
| Education status of respondents | 0.000 |
| Farm land size (hectare) | 0.000 |
| Frequency of crop failure in 30 years (1987–2016) | 0.000 |
| Frequency of drought/flood in 30 years (1987–20160) | 0.000 |
| Irrigated land in hector | 0.005 |
| Perception of respondents on climate change | 0.000 |
| Production of sorghum in quintals (main crop) | 0.272 |
| Soil fertility status of farm land | 0.000 |
| Standard score (Z-score) of animal resources | 0.000 |

Degree of freedom (df) = 2.

0.7078. Soil fertility status of the area is measured by the perception of smallholder farmers. Soil fertility is rated as high, medium and low. This is an important component of agricultural adaptation planning. This was used by [4] in their study of vulnerability of smallholder farmers to climate change and variability in the central highlands of Ethiopia.

*Natural resources.* Sub-components of natural resource are: HHs exploit natural resources (43% in *Kolla*, 26% in *Woyinadaga*, and 18% in *Daga*), HHs using only forest-based energy for cooking (*Kolla* 86%, *Woyinadaga* 74%, *Daga* 26%), HHs reporting that firewood is decreasing in the last 30 years (*Kolla* 98%, *Woyinadaga* 100%, *Daga* 52%).Moreover,HHs reporting land degradation by climate change (*Kolla* 97%, *Woyinadaga* 99%, *Daga* 79%). The profile index values are 0.82, 0.75, and 0.44 in *Kolla*, *Woyinadaga*, and *Daga* respectively (Table 6). Consequently, *Kolla* (0.82) was vulnerable in terms of unwise use of natural resources and deforestation for home energy while *Daga* agro-ecology is less vulnerable. A similar result was reported by [8].

Overall, LVI result revealed 0.18 of *Kolla* which is most vulnerable while *Daga* is least vulnerable (0.08), and *Woyinadaga* is moderate (0.13) in the study area (Table 8). The result also

**Table 8. LVI/ LVI-IPCC contributing factors across agro-ecology.**

| Vulnerability components | Kolla | Woyina Daga | Daga |
|---|---|---|---|
| AC | 0.49 | 0.41 | 0.43 |
| Exposure | 0.74 | 0.7 | 0.61 |
| Sensitivity | 0.71 | 0.46 | 0.42 |
| LVI | 0.65 | 0.52 | 0.49 |
| LVI-IPCC | 0.18 | 0.13 | 0.08 |

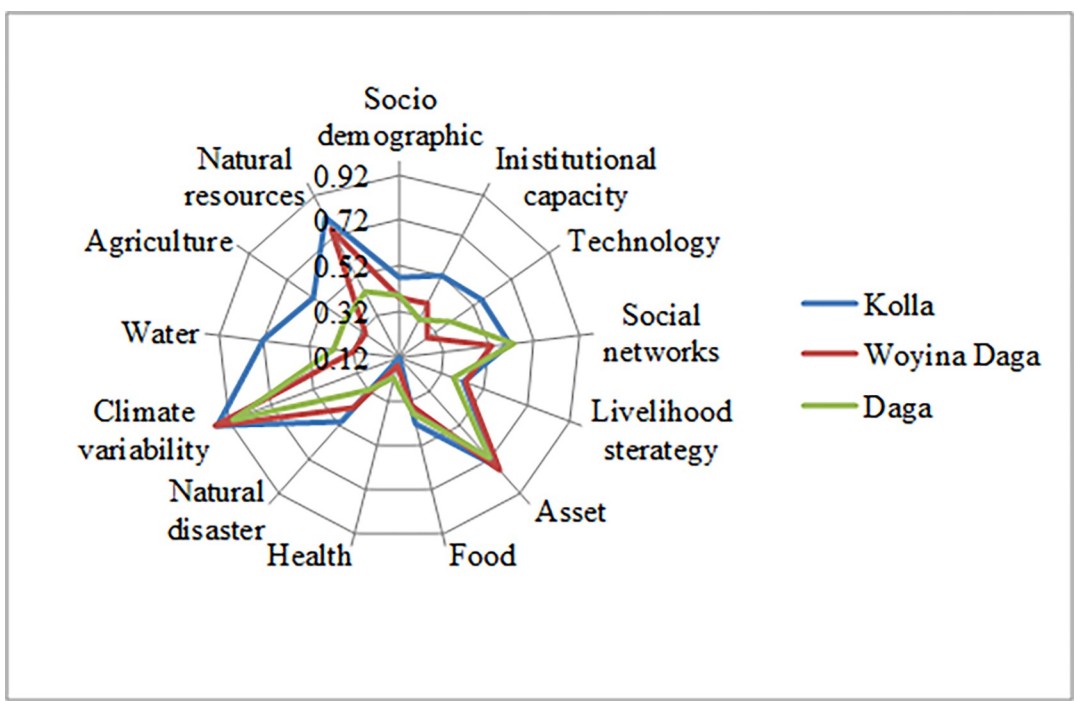

**Fig 2. Vulnerability Spider Diagram of the major components of the Livelihood Vulnerability Index (LVI) for the study area.**

aligned with the discussion made with experts as well as the *Kebeles'* key informant interviewee of the three agro-ecologies. They uniformly stated that the vulnerability of Kolla is higher than other AEZs.This is inline with the previous studies [10,23,24] who revealed that *Kolla* agro-ecology is found the most vulnerable. However, the result disagree with the study by [20] who found that highland (*Dega*) is the most vulnerable while the lowland (*Kolla*) is not vulnerable.

(Fig 2) presents 13 major components. The scale of the diagram goes from the center 0.2 (less vulnerable) to 0.98 (more vulnerable). It indicates *Kolla* agro-ecology is more vulnerable in terms of socio-demographic, institutional capacity, technology, food, natural disaster, water, agriculture, and natural resources. Besides, *Woyinadaga* is negatively influenced by livelihood strategy, asset, and climate variability while *Daga* is solely vulnerable in terms of the social network.

## Livelihood Vulnerability Index (LVI)–IPCC results

Table 8 presents the three contributing factors to climate change vulnerability-exposure, sensitivity, and adaptive capacity that differ across the three agro-ecologies. As it is evident from the equation for IPCC_LVI, high values of exposure relative to adaptive capacity assume positive vulnerability scores while low values of exposure relative to adaptive capacity yield negative vulnerability scores. Sensitivity acts as a multiplier, such that high sensitivity in an agro-ecology for which exposure exceeds adaptive capacity will result in a larger positive LVI-IPCC vulnerability scores [18].

As Fig 3 and Table 8 depict, *Kolla* agro-ecology has highest exposure (0.74) with low adaptive capacity (0.49) followed by *Woyinadaga* agro-ecology with a score of 0.7 exposure, 0.46 sensitivity, and 0.41 adaptive capacity. *Daga* has 0.43 adaptive capacities, 0.61 exposures, and 0.42 sensitivities. Daga has lowest exposure compared to others which makes it the least vulnerable.

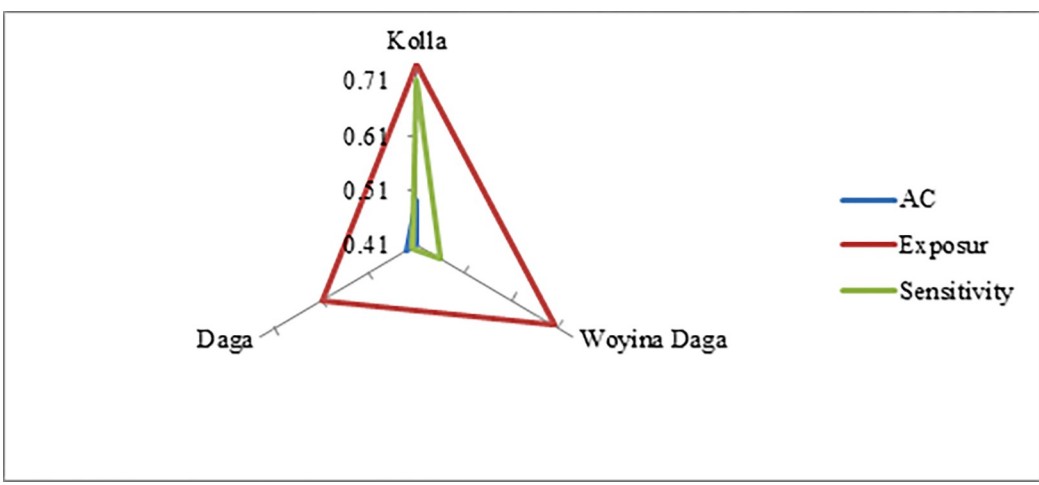

**Fig 3. Vulnerability triangles of LVI-IPCC contributing factors.**

Comprehensively, the LVI–IPCC result revealed *Kolla* agro-ecology is most vulnerable due to exposure and sensitivity of *Kolla* exceeding adaptive capacity. *Daga* is least vulnerable because of its lowest exposure and sensitivity compared to others and better adaptive capacity though not exceeds Kolla's adaptive capacity. *Woyinadaga* is moderately vulnerable. The result was inline with the previous study [8] who revealed that the highest exposure and sensitivity coupled with lowest adaptive capacity in *Kolla* made it the most vulnerable; Dega agro-ecology is least vulnerable.

In general, the two methods resulted in similar degrees of vulnerability and identified Kolla agro-ecological zone as the most vulnerable while the Dega agro-ecological zone is the least vulnerable of the three agro-ecological zones. This household-level assessment of vulnerability is useful to identify and prioritize vulnerable areas and contributing factors of vulnerability. Our results corroborate the findings of [3,5] who reported Kolla agro-ecological zone is most vulnerable than others.

Our results identify Oromo Nationality Administration is vulnerable to climate change and variability. Current climate variability has a significant influence on the agriculture sector. Moreover, water shortage is a critical challenge in the area. Any unfavorable change in the local climate in the future will have serious implications for household level food security. The findings have implications for planning and prioritizing adaptation interventions in the study area by identifying and prioritizing vulnerable areas to climate change and variability.

## Conclusion

This study examines agro-ecological based smallholder farmers' livelihood vulnerability to climate variability and change in Oromo Nationality Administration, north-east Ethiopia. LVI result reveled *Kolla* (0.18) which is most vulnerable while *Daga* is least vulnerable (0.08), and *Woyinadaga* is moderate (0.13) in Oromo Nationality Administration. Akin, *Kolla* agro-ecology is more vulnerable in terms of socio-demographic, institutional capacity, technology, food, natural disaster, water, agriculture, and natural resources. Besides, *Woyinadaga* is negatively influenced by livelihood strategy, asset, and climate variability. Daga is solely vulnerable in terms of social network. Similarly, LVI–IPCC result revealed that Kolla agro-ecology is most vulnerable as exposure scores of *Kolla* exceeds its adaptive capacity. *Daga* is least vulnerable because of its lowest exposure and better adaptive capacity; *Woyinadaga* is moderately vulnerable.

It is imperative to give closer attention to planning adaptation options to reduce current and future vulnerability based on agro-ecology and socio-economic context. Therefore, in the study area in general and in Kolla agro-ecology in particular plannedadaptation is needed. In addition, future research needs to investigate adaptation strategy and factors affecting it, assessment of household food security in the face of climate change, and variability to fully realize the extent of vulnerability and implication to further forward policy implications.

## Supporting information

**S1 Data.**
(SAV)

**S1 File.**
(DOCX)

## Acknowledgments

Thanks to the interviewees and the farmers of the study area for contributing their share for the fruitfulness of this research. The paper has benefited from useful comments and suggestions by two anonymous referees to whom the authors are very grateful.

## Author Contributions

**Conceptualization:** Ahmed Aliy Ebrahim, Birhan Asmame Miheretu, Arragaw Alemayehu.

**Data curation:** Ahmed Aliy Ebrahim, Birhan Asmame Miheretu.

**Formal analysis:** Ahmed Aliy Ebrahim, Birhan Asmame Miheretu, Arragaw Alemayehu.

**Funding acquisition:** Ahmed Aliy Ebrahim.

**Investigation:** Ahmed Aliy Ebrahim, Birhan Asmame Miheretu, Arragaw Alemayehu.

**Methodology:** Ahmed Aliy Ebrahim, Birhan Asmame Miheretu, Arragaw Alemayehu.

**Project administration:** Ahmed Aliy Ebrahim, Birhan Asmame Miheretu.

**Resources:** Ahmed Aliy Ebrahim, Birhan Asmame Miheretu.

**Software:** Ahmed Aliy Ebrahim, Birhan Asmame Miheretu.

**Supervision:** Ahmed Aliy Ebrahim, Birhan Asmame Miheretu.

**Validation:** Ahmed Aliy Ebrahim, Birhan Asmame Miheretu, Arragaw Alemayehu.

**Visualization:** Ahmed Aliy Ebrahim, Birhan Asmame Miheretu, Arragaw Alemayehu.

**Writing – original draft:** Ahmed Aliy Ebrahim, Birhan Asmame Miheretu.

**Writing – review & editing:** Ahmed Aliy Ebrahim, Birhan Asmame Miheretu, Arragaw Alemayehu.

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
