## [Decision Letter · Decision Letter 0]

11 Oct 2021

PONE-D-21-29744Vulnerability of small holder farmers to climate variability and change across different agro-ecological Zones in Oromo Nationality Administration (ONA), North east EthiopiaPLOS ONE

Dear Dr. Ebrahim,

Thank you for submitting your manuscript to PLOS ONE. After careful consideration, we feel that it has merit but does not fully meet PLOS ONE’s publication criteria as it currently stands. Therefore, we invite you to submit a revised version of the manuscript that addresses the points raised during the review process.

We look forward to receiving your revised manuscript.

Kind regards,

Shamsuddin Shahid

Academic Editor

PLOS ONE

Journal Requirements:

When submitting your revision, we need you to address these additional requirements. 1. Please ensure that your manuscript meets PLOS ONE's style requirements, including those for file naming. The PLOS ONE style templates can be found at https://journals.plos.org/plosone/s/file?id=wjVg/PLOSOne_formatting_sample_main_body.pdf and https://journals.plos.org/plosone/s/file?id=ba62/PLOSOne_formatting_sample_title_authors_affiliations.pdf 2. Please include additional information regarding the survey or questionnaire used in the study and ensure that you have provided sufficient details that others could replicate the analyses. For instance, if you developed a questionnaire as part of this study and it is not under a copyright more restrictive than CC-BY, please include a copy, in both the original language and English, as Supporting Information. If the original language is written in non-Latin characters, for example Amharic, Chinese, or Korean, please use a file format that ensures these characters are visible. 3. Please include your tables as part of your main manuscript and remove the individual files. Please note that supplementary tables (should remain/ be uploaded) as separate "supporting information" files.’ 4. We note that you have indicated that data from this study are available upon request. PLOS only allows data to be available upon request if there are legal or ethical restrictions on sharing data publicly. For more information on unacceptable data access restrictions, please see http://journals.plos.org/plosone/s/data-availability#loc-unacceptable-data-access-restrictions.  In your revised cover letter, please address the following prompts: a) If there are ethical or legal restrictions on sharing a de-identified data set, please explain them in detail (e.g., data contain potentially sensitive information, data are owned by a third-party organization, etc.) and who has imposed them (e.g., an ethics committee). Please also provide contact information for a data access committee, ethics committee, or other institutional body to which data requests may be sent. b) If there are no restrictions, please upload the minimal anonymized data set necessary to replicate your study findings as either Supporting Information files or to a stable, public repository and provide us with the relevant URLs, DOIs, or accession numbers. For a list of acceptable repositories, please see http://journals.plos.org/plosone/s/data-availability#loc-recommended-repositories. We will update your Data Availability statement on your behalf to reflect the information you provide. 5. Please include your full ethics statement in the ‘Methods’ section of your manuscript file. In your statement, please include the full name of the IRB or ethics committee who approved or waived your study, as well as whether or not you obtained informed written or verbal consent. If consent was waived for your study, please include this information in your statement as well.  6. Please ensure that you refer to Figure 1 in your text as, if accepted, production will need this reference to link the reader to the figure. 7. We note that Figure 1 in your submission contain [map/satellite] images which may be copyrighted. All PLOS content is published under the Creative Commons Attribution License (CC BY 4.0), which means that the manuscript, images, and Supporting Information files will be freely available online, and any third party is permitted to access, download, copy, distribute, and use these materials in any way, even commercially, with proper attribution. For these reasons, we cannot publish previously copyrighted maps or satellite images created using proprietary data, such as Google software (Google Maps, Street View, and Earth). For more information, see our copyright guidelines: http://journals.plos.org/plosone/s/licenses-and-copyright. We require you to either (1) present written permission from the copyright holder to publish these figures specifically under the CC BY 4.0 license, or (2) remove the figures from your submission: a. You may seek permission from the original copyright holder of Figure 1 to publish the content specifically under the CC BY 4.0 license.   We recommend that you contact the original copyright holder with the Content Permission Form (http://journals.plos.org/plosone/s/file?id=7c09/content-permission-form.pdf) and the following text:“I request permission for the open-access journal PLOS ONE to publish XXX under the Creative Commons Attribution License (CCAL) CC BY 4.0 (http://creativecommons.org/licenses/by/4.0/). Please be aware that this license allows unrestricted use and distribution, even commercially, by third parties. Please reply and provide explicit written permission to publish XXX under a CC BY license and complete the attached form.” Please upload the completed Content Permission Form or other proof of granted permissions as an "Other" file with your submission. In the figure caption of the copyrighted figure, please include the following text: “Reprinted from [ref] under a CC BY license, with permission from [name of publisher], original copyright [original copyright year].” b. If you are unable to obtain permission from the original copyright holder to publish these figures under the CC BY 4.0 license or if the copyright holder’s requirements are incompatible with the CC BY 4.0 license, please either i) remove the figure or ii) supply a replacement figure that complies with the CC BY 4.0 license. Please check copyright information on all replacement figures and update the figure caption with source information. If applicable, please specify in the figure caption text when a figure is similar but not identical to the original image and is therefore for illustrative purposes only.The following resources for replacing copyrighted map figures may be helpful: USGS National Map Viewer (public domain): http://viewer.nationalmap.gov/viewer/The Gateway to Astronaut Photography of Earth (public domain): http://eol.jsc.nasa.gov/sseop/clickmap/Maps at the CIA (public domain): https://www.cia.gov/library/publications/the-world-factbook/index.html and https://www.cia.gov/library/publications/cia-maps-publications/index.htmlNASA Earth Observatory (public domain): http://earthobservatory.nasa.gov/Landsat: http://landsat.visibleearth.nasa.gov/USGS EROS (Earth Resources Observatory and Science (EROS) Center) (public domain): http://eros.usgs.gov/#Natural Earth (public domain): http://www.naturalearthdata.com/ 8. We note you have included a table to which you do not refer in the text of your manuscript. Please ensure that you refer to Table 3 in your text; if accepted, production will need this reference to link the reader to the Table.

Reviewers' comments:

Reviewer's Responses to Questions

**Comments to the Author**

1. Is the manuscript technically sound, and do the data support the conclusions?

Reviewer #1: Partly

Reviewer #2: Yes

2. Has the statistical analysis been performed appropriately and rigorously? 

Reviewer #1: N/A

Reviewer #2: I Don't Know

3. Have the authors made all data underlying the findings in their manuscript fully available?

Reviewer #1: Yes

Reviewer #2: Yes

4. Is the manuscript presented in an intelligible fashion and written in standard English?

Reviewer #1: Yes

Reviewer #2: Yes

5. Review Comments to the Author

Reviewer #1: The study tried to assess vulnerability of smallholder farmers to climate variability and change across different agro-ecological zones in Oromo Nationality Administration (ONA), North east of Ethiopia. I would like to recommend the authors to improve the manuscript considering the following comments.

What is the novelty of study? There is a body of literature conducted to assess vulnerability of smallholder farmers to climate variability based on agro-ecological system approach, over Ethiopia (e.g. Asfaw et al,. 2021, Gebru at al., 2020, Mohammed et al., 2021, Zeleke at al., 2021) and a few of the research works have been also cited in manuscript (ref. number: 7,8, 23 & 24). What is the particular problem or approach that nobody else tried for it yet? The authors claim “little studies used expert judgment approach in weighting indicators of vulnerability components. Hence, this study aimed to fill the gap in methodology.” However, the authors did not also provide a novel method for their analysis, and simply applied the methods commonly employed in the published works.

In addition to using only expert judgment for assigning weight to indicators, the study could use statistical method (Principal Component Analysis (PCA) or Factor Analysis (FA)) or the arbitrary choice of equal weight. Though methods have been extensively applied in the literature, the comparison can add new findings to the existing literature.

This study appears to be site-specific and can be more considered as a case study which may be interesting for local communities but not for international audiences. Moreover, it is difficult for readers to understand and follow information under LVI results section. Therefore, it is helpful to define levels of vulnerability to each agro-ecology. Authors are suggested to consider vulnerability scores to categorize vulnerability across the study area with low vulnerable, moderate and high vulnerable levels.

The manuscript needs a round of proofreading. Many sentences should be completed by dot after reference number. Grammar and English writing need to be improved. For example “Little studies” should be revised into “few studies”.

The other concern is related to some apparent missing in manuscript. It is suggested that authors review the manuscript carefully. Examples are as below:

a) Where is the parameter “n” in equation (1)? (It should be clear for reader that equation (1) is equal to n which is the total sample size for the study). Moreover, a part of Equation 1; (N-2), differs from the calculation four lines after (6347-1), and It seems this part should be edited to (N-1) in the equation as below:

n= (z^2*p*q*N )/(e^2 (N-1)+Z^2*p*q)

b) Reference 24: the referencing is wrong, it seems that the authors mixed two following reference together!

Chala, D., Belay, S., Bamlaku, A., & Azadi, H. (2016). Agro-ecological based small-holder farmer's livelihoods vulnerability to climate variability and change in Didesa sub Basin of Blue Nile River, Ethiopia. Academia Journal of Agricultural Research, 4(5), 230-240.

Dechassa, C., Simane, B., & Alamirew, B. (2017). Farmers’ livelihoods vulnerability to climate variability and change in Didesa Basin southern part of Abay Basin, Ethiopia. In Climate Change Adaptation in Africa (pp. 267-284). Springer, Cham.

c) In addition to “Livelihood Vulnerability Index (LVI)” Authors should make it clear for reader that “LVI-IPCC” is abbreviation for LVI-Intergovernmental Panel on Climate Change.

References in review:

Asfaw, A., Bantider, A., Simane, B., & Hassen, A. (2021). Smallholder farmers’ livelihood vulnerability to climate change-induced hazards: agroecology-based comparative analysis in Northcentral Ethiopia (Woleka Sub-basin). Heliyon, 7(4), e06761.

Gebru, G. W., Ichoku, H. E., & Phil-Eze, P. O. (2020). Determinants of smallholder farmers' adoption of adaptation strategies to climate change in Eastern Tigray National Regional State of Ethiopia. Heliyon, 6(7), e04356.

Mohammed, Y., Tesfaye, K., Tadesse, M., & Yimer, F. (2021). Analysis of smallholder farmers’ vulnerability to climate change and variability in south Wollo, north east highlands of Ethiopia: An agro-ecological system-based approach.

Zeleke, T., Beyene, F., Deressa, T., Yousuf, J., & Kebede, T. (2021). Vulnerability of Smallholder Farmers to Climate Change-Induced Shocks in East Hararghe Zone, Ethiopia. Sustainability 2021, 13, 2162.

Reviewer #2: 1)Justification of the study should be more elaborated. Why was this area selected as the study location? Why only smallholder farmers had been considered? - these issues should be addressed clearly.

2)Methodology should be more explained. Especially, variables/indicators used in the study could be summarized in a table with adequate information.

3)Justification of indicator/variable selection along with references could improve this section more.

4)Why only 3 indicators were considered for ‘Technology’ component? - Explain with reference.

5)Why only three indicators were considered for Asset? How you have measured the ‘fertile soil of the farm land’? Please explain with necessary references. Try to compare the findings with previous studies.

6)How ‘enough food for nutrition’ mentioned under ‘Food’ sub-section in result and discussion section, was measured? It should be mentioned in methodology. Compare the result of this sub-section with previous studies. Please make some discussions based on your findings instead of just describing the results.

7)‘Health’ sub-section under result and discussion part should be re-written by making it more clear to the readers.

8)In case of frequency of flood and drought, you are mentioning in text about last 10 years but in parenthesis you have written 2011-2016. Please make correction.

9)In case of climate variability change, it would be better to include secondary climate data and compare with farmers perception.

10) Please compare the findings of ‘Water’ sub-section with previous findings.

11) What are the reasons behind the perception of decrease in crop yield mentioned in the ‘Agriculture’ sub-section, please mention?

12) More effort should be made on the implication of the study findings in ‘result and discussion’ section.

13) For indicating the citation in the main text, it is appropriate to use […] instead of (…).

14) There are some typo throughout the manuscript. Please go through carefully and make

6. PLOS authors have the option to publish the peer review history of their article (what does this mean?). If published, this will include your full peer review and any attached files.

Reviewer #1: No

Reviewer #2: No

---

## [Author Response · Author response to Decision Letter 0]

14 Mar 2022

Dear, Shamsuddin Shahid

Academic Editor

PLOS ONE

Subject: Ref.: No. PONE-D-21-29744

We are writing this with reference to the revised manuscript entitled Vulnerability of small holder farmers to climate variability and change across different agro-ecological Zones in Oromo Nationality Administration (ONA), North east Ethiopia, which has be submitted for publication in PLOS ONE. We would like to express our sincere gratitude to the reviewers for their valuable comments and suggestions. We have revised the manuscript by taking into account the comments and suggestions given by reviewers (Reviewers #1 and #2). We are now re-submitting the revised version for your kind reconsideration for publication. We sincerely hope that we have sufficiently addressed the suggested comments, and have prepared the manuscript in accordance with the journal’s style. The details of how the comments and suggestions as addressed point by point are given below. 

Looking forward to hear your positive response to publish our manuscript in your Journal, 

=====

Rebuttal letter

Response to the Journal Requirements

Response: We thank you for your key comments and we revised the manuscript based on POLS ONE manuscript preparation templates including file naming (Please see the revised version).

2. Please include additional information regarding the survey or questionnaire used in the study and ensure that you have provided sufficient details that others could replicate the analyses. For instance, if you developed a questionnaire as part of this study and it is not under a copyright more restrictive than CC-BY, please include a copy, in both the original language and English, as Supporting Information. If the original language is written in non-Latin characters, for example Amharic, Chinese, or Korean, please use a file format that ensures these characters are visible.

Response: Thank you. We have uploaded the survey or questionnaire in the supporting information section.

3. Please include your tables as part of your main manuscript and remove the individual files. Please note that supplementary tables (should remain/ be uploaded) as separate "supporting information" files.’

Response: The tables are included in the main manuscript.

Response: the minimal data was uploaded in the supporting information section.

6. Please ensure that you refer to Figure 1 in your text as, if accepted, production will need this reference to link the reader to the figure.

Response: Thank you. We have refer figure 1 in the revised manuscript (Please see revised manuscript with track changes document on page 5 line 128.

7. We note that Figure 1 in your submission contain [map/satellite] images which may be copyrighted. All PLOS content is published under the Creative Commons Attribution License (CC BY 4.0), which means that the manuscript, images, and Supporting Information files will be freely available online, and any third party is permitted to access, download, copy, distribute, and use these materials in any way, even commercially, with proper attribution. For these reasons, we cannot publish previously copyrighted maps or satellite images created using proprietary data, such as Google software (Google Maps, Street View, and Earth). For more information, see our copyright guidelines: http://journals.plos.org/plosone/s/licenses-and-copyright.

Response: The map is prepared by principal investigator Ahmed Aliy Ebrahim using ARCGIS 10.3 

8. We note you have included a table to which you do not refer in the text of your manuscript. Please ensure that you refer to Table 3 in your text; if accepted, production will need this reference to link the reader to the Table.

Response: Thank you. We have referred table 3 in the revised manuscript (Please see revised manuscript with track changes document on page 14 line 351).

Reviewers' comments:

Reviewer's Responses to Questions

Comments to the Author

1. Is the manuscript technically sound, and do the data support the conclusions?

Reviewer #1: Partly

Reviewer #2: Yes

Response: Thank you very much for recognizing our study importance. We have modified the manuscript in accordance with the reviewers comment.

2. Has the statistical analysis been performed appropriately and rigorously?

Reviewer #1: N/A

Reviewer #2: I Don't Know

Response: Thank you for the critical observation of statistical analysis. We have improved the statistical analysis procedures and interpretations in the revised manuscript based on the reviewers comment.

3. Have the authors made all data underlying the findings in their manuscript fully available?

Reviewer #1: Yes

Reviewer #2: Yes

Response: Thank you..

4. Is the manuscript presented in an intelligible fashion and written in standard English?

Reviewer #1: Yes

Reviewer #2: Yes

Response: Thank you for the positive remark on our manuscript and we really appreciate your scientific judgment.

5. Review Comments to the Author

Response: Thank you very much for your concern.

Reviewer #1: The study tried to assess vulnerability of smallholder farmers to climate variability and change across different agro-ecological zones in Oromo Nationality Administration (ONA), North east of Ethiopia. I would like to recommend the authors to improve the manuscript considering the following comments.

What is the novelty of study? There is a body of literature conducted to assess vulnerability of smallholder farmers to climate variability based on agro-ecological system approach, over Ethiopia (e.g. Asfaw et al,. 2021, Gebru at al., 2020, Mohammed et al., 2021, Zeleke at al., 2021) and a few of the research works have been also cited in manuscript (ref. number: 7,8, 23 & 24). 

Response: Thank you for your constructive comments and suggestions. We have added the innovative element of the present study. (Please see line number 106-126 on page 4 and 5 in revised manuscript with track changes word document)

What is the particular problem or approach that nobody else tried for it yet? The authors claim “little studies used expert judgment approach in weighting indicators of vulnerability components. Hence, this study aimed to fill the gap in methodology.” However, the authors did not also provide a novel method for their analysis, and simply applied the methods commonly employed in the published works.

Response: The methodological contribution is corrected as the methodology is applicable to other parts of Ethiopia as well.

In addition to using only expert judgment for assigning weight to indicators, the study could use statistical method (Principal Component Analysis (PCA) or Factor Analysis (FA)) or the arbitrary choice of equal weight. Though methods have been extensively applied in the literature, the comparison can add new findings to the existing literature.

Response: A brief description on the technique used is included. The methodology section was more elaborated.

This study appears to be site-specific and can be more considered as a case study which may be interesting for local communities but not for international audiences. Moreover, it is difficult for readers to understand and follow information under LVI results section. Therefore, it is helpful to define levels of vulnerability to each agro-ecology. Authors are suggested to consider vulnerability scores to categorize vulnerability across the study area with low vulnerable, moderate and high vulnerable levels.

Response: Thank you for your comments. The vulnerability analysis is disaggregated at agro-ecological level and each agro-ecology is categorized as high, moderate and low levels of vulnerability depending on the scores obtained.

The manuscript needs a round of proofreading. Many sentences should be completed by dot after reference number. Grammar and English writing need to be improved. For example “Little studies” should be revised into “few studies”.

Response: Thank you for this comment and we have addressed the issue of language editing. The manuscript is reviewed and edited by English language editor (professional). Hence, we believe that the language of the current version of the paper is significantly improved and consequently the quality of the paper too. (You can see revised manuscript with track changes document).

The other concern is related to some apparent missing in manuscript. It is suggested that authors review the manuscript carefully. Examples are as below:

a) Where is the parameter “n” in equation (1)? (It should be clear for reader that equation (1) is equal to n which is the total sample size for the study). Moreover, a part of Equation 1; (N-2), differs from the calculation four lines after (6347-1), and It seems this part should be edited to (N-1) in the equation as below:

n= (z^2*p*q*N )/(e^2 (N-1)+Z^2*p*q)

Response: Fixed.

b) Reference 24: the referencing is wrong, it seems that the authors mixed two following reference together!

Chala, D., Belay, S., Bamlaku, A., & Azadi, H. (2016). Agro-ecological based small-holder farmer's livelihoods vulnerability to climate variability and change in Didesa sub Basin of Blue Nile River, Ethiopia. Academia Journal of Agricultural Research, 4(5), 230-240.

Dechassa, C., Simane, B., & Alamirew, B. (2017). Farmers’ livelihoods vulnerability to climate variability and change in Didesa Basin southern part of Abay Basin, Ethiopia. In Climate Change Adaptation in Africa (pp. 267-284). Springer, Cham.

c) In addition to “Livelihood Vulnerability Index (LVI)” Authors should make it clear for reader that “LVI-IPCC” is abbreviation for LVI-Intergovernmental Panel on Climate Change.

Response: All missed elements are now addressed.

Reviewer #2: 1)Justification of the study should be more elaborated. Why was this area selected as the study location? Why only smallholder farmers had been considered? - these issues should be addressed clearly.

Response: Thank you for your scientific comments. A sound justification is added. 

2)Methodology should be more explained. Especially, variables/indicators used in the study could be summarized in a table with adequate information.

Response: Detailed description is included in the methodology section.

3)Justification of indicator/variable selection along with references could improve this section more.

Response: The indicator/variable selection section is more elaborated and presented with reference.

4)Why only 3 indicators were considered for ‘Technology’ component? - Explain with reference.

Response: A sound justification is added on why the present study is focused on 3 indicators for ‘technology components. Please see line number 461-464

5)Why only three indicators were considered for Asset? 

Response: A sound justification is added on why the present study is focused on 3 indicators for asset’ components. Please see line number 491-495.

How you have measured the ‘fertile soil of the farm land’? Please explain with necessary references. Try to compare the findings with previous studies

. Response: Fixed. Please see line number 586-589.

6)How ‘enough food for nutrition’ mentioned under ‘Food’ sub-section in result and discussion section, was measured? It should be mentioned in methodology. Compare the result of this sub-section with previous studies. Please make some discussions based on your findings instead of just describing the results.

Response: ‘Enough food for nutrition’ is corrected as enough throughout the year

7)‘Health’ sub-section under result and discussion part should be re-written by making it more clear to the readers.

Response: The ‘health’ section is re-written to make it more clear to the readers.

8)In case of frequency of flood and drought, you are mentioning in text about last 10 years but in parenthesis you have written 2011-2016. Please make correction.

Response: The inconsistency is edited through the manuscript and corrected as 1987-2016.

9)In case of climate variability change, it would be better to include secondary climate data and compare with farmers perception.

Response: Results of secondary data on rainfall and temperature was included and compared with perceptions of farmers in a separate section.

10) Please compare the findings of ‘Water’ sub-section with previous findings.

Response: Results in the ‘Water’ sub-section are compared with previous findings.

11) What are the reasons behind the perception of decrease in crop yield mentioned in the ‘Agriculture’ sub-section, please mention?

Response: Reasons for perception of decrease in crop yield should are added in the results and discussion section.

12) More effort should be made on the implication of the study findings in ‘result and discussion’ section.

Response: The implication of the findings of the present study is included at the end of the results and discussion section.

13) For indicating the citation in the main text, it is appropriate to use […] instead of (…).

Response: Fixed.

14) There are some typo throughout the manuscript. Please go through carefully and make

Response: Thank you for this comment and we addressed the issue of language editing. Dear reviewer, we thank you for these key comments and we agreed with all your ideas and the manuscript is already updated accordingly. and please see revised manuscript with track changes document.

---

## [Decision Letter · Decision Letter 1]

4 Apr 2022

PONE-D-21-29744R1Vulnerability of small holder farmers to climate variability and change across different agro-ecological Zones in Oromo Nationality Administration (ONA), North east EthiopiaPLOS ONE

Dear Dr. Ebrahim,

Thank you for submitting your manuscript to PLOS ONE. After careful consideration, we feel that it has merit but does not fully meet PLOS ONE’s publication criteria as it currently stands. Therefore, we invite you to submit a revised version of the manuscript that addresses the points raised during the review process.

We look forward to receiving your revised manuscript.

Kind regards,

Shamsuddin Shahid

Academic Editor

PLOS ONE

Journal Requirements:

Reviewers' comments:

Reviewer's Responses to Questions

**Comments to the Author**

1. If the authors have adequately addressed your comments raised in a previous round of review and you feel that this manuscript is now acceptable for publication, you may indicate that here to bypass the “Comments to the Author” section, enter your conflict of interest statement in the “Confidential to Editor” section, and submit your "Accept" recommendation.

Reviewer #1: (No Response)

Reviewer #2: All comments have been addressed

2. Is the manuscript technically sound, and do the data support the conclusions?

Reviewer #1: Yes

Reviewer #2: Yes

3. Has the statistical analysis been performed appropriately and rigorously? 

Reviewer #1: N/A

Reviewer #2: Yes

4. Have the authors made all data underlying the findings in their manuscript fully available?

Reviewer #1: Yes

Reviewer #2: Yes

5. Is the manuscript presented in an intelligible fashion and written in standard English?

Reviewer #1: Yes

Reviewer #2: Yes

6. Review Comments to the Author

Reviewer #1: The authors have adequately addressed the comments raised in the first round of review. The manuscript is more coherent now and the quality of the revised manuscript has been improved. Overall, the paper is well written and sounds technically good.

There are minor comments before considering for publication.

Abstract, Line 53: It is suggested to avoid using the personal form (we conclude …). Passive voice and third person form should be used for the whole manuscript.

Keywords, agro-ecology: Please capitalize first word

Introduction, line 112: “As a result, this paper aimed to examine agro-ecological based…”

&

Conclusion, line 667: “This study examines agro ecological based ...”,

Please be consistent for using hyphen over the whole paper and it looks it can be edited as “… agro-ecological zones…” to better conveying of meaning.

Reviewer #2: (No Response)

7. PLOS authors have the option to publish the peer review history of their article (what does this mean?). If published, this will include your full peer review and any attached files.

Reviewer #1: No

Reviewer #2: **Yes: **Mohammad Kamruzzaman, Farm Machinery and Postharvest Technology Division, Bangladesh Rice Research Institute, Gazipur-1701, Bangladesh, email: milonbrri@gmail.com

---

## [Author Response · Author response to Decision Letter 1]

20 Apr 2022

Dear, Shamsuddin Shahid

Academic Editor

PLOS ONE

Subject: Ref.: No. PONE-D-21-29744R1

We are writing this with reference to the revised manuscript entitled Vulnerability of small holder farmers to climate variability and change across different agro-ecological Zones in Oromo Nationality Administration (ONA), North east Ethiopia, which has be submitted for publication in PLOS ONE. We would like to express our sincere gratitude to the reviewers for their valuable comments and suggestions. We have revised the manuscript by taking into account the comments and suggestions given by reviewers (Reviewers #1 and #2). We are now re-submitting the revised version for your kind reconsideration for publication. We sincerely hope that we have sufficiently addressed the suggested comments, and have prepared the manuscript in accordance with the journal’s style. The details of how the comments and suggestions as addressed point by point are given below. 

Looking forward to hear your positive response to publish our manuscript in your Journal, 

=====

Rebuttal letter

Response to the Journal Requirements

Response: The reference list is complete and correct.

Reviewers' comments:

Reviewer's Responses to Questions

Comments to the Author

1. If the authors have adequately addressed your comments raised in a previous round of review and you feel that this manuscript is now acceptable for publication, you may indicate that here to bypass the “Comments to the Author” section, enter your conflict of interest statement in the “Confidential to Editor” section, and submit your "Accept" recommendation.

Reviewer #1: (No Response)

Reviewer #2: All comments have been addressed

Response: Thank you very much

2. Is the manuscript technically sound, and do the data support the conclusions?

Reviewer #1: Yes

Reviewer #2: Yes

Response: Thank you very much

3. Has the statistical analysis been performed appropriately and rigorously?

Reviewer #1: N/A

Reviewer #2: Yes

Response: Thank you very much for your comments.

4. Have the authors made all data underlying the findings in their manuscript fully available?

Reviewer #1: Yes

Reviewer #2: Yes

Response: Thank you very much

5. Is the manuscript presented in an intelligible fashion and written in standard English?

Reviewer #1: Yes

Reviewer #2: Yes

Response: Thank you very much for your comments.

6. Review Comments to the Author

Reviewer #1: The authors have adequately addressed the comments raised in the first round of review. The manuscript is more coherent now and the quality of the revised manuscript has been improved. Overall, the paper is well written and sounds technically good.

Response: Thank you very much for your positive response

There are minor comments before considering for publication.

Abstract, Line 53: It is suggested to avoid using the personal form (we conclude …). Passive voice and third person form should be used for the whole manuscript.

Response: Fixed. Thank you very much for your comments.

Keywords, agro-ecology: Please capitalize first word

Response: Fixed.

Introduction, line 112: “As a result, this paper aimed to examine agro-ecological based…”

&

Conclusion, line 667: “This study examines agro ecological based ...”,

Please be consistent for using hyphen over the whole paper and it looks it can be edited as “… agro-ecological zones…” to better conveying of meaning.

Response: Fixed. Thank you very much for your comments.

Reviewer #2: (No Response)

7. PLOS authors have the option to publish the peer review history of their article (what does this mean?). If published, this will include your full peer review and any attached files.

Do you want your identity to be public for this peer review? For information about this choice, including consent withdrawal, please see our Privacy Policy.

Reviewer #1: No

Reviewer #2: Yes: Mohammad Kamruzzaman, Farm Machinery and Postharvest Technology Division, Bangladesh Rice Research Institute, Gazipur-1701, Bangladesh, email: milonbrri@gmail.com

---

## [Editor Report · Decision Letter 2]

22 Apr 2022

Vulnerability of small holder farmers to climate variability and change across different agro-ecological Zones in Oromo Nationality Administration (ONA), North east Ethiopia

PONE-D-21-29744R2

Dear Dr. Ebrahim,

We’re pleased to inform you that your manuscript has been judged scientifically suitable for publication and will be formally accepted for publication once it meets all outstanding technical requirements.

Kind regards,

Shamsuddin Shahid

Academic Editor

PLOS ONE
---

## [Editor Report · Acceptance letter]

20 May 2022

PONE-D-21-29744R2 

Vulnerability of smallholder farmers to climate variability and change across different agro-ecological Zones in Oromo Nationality Administration (ONA), North east Ethiopia 

Dear Dr. Ebrahim:

I'm pleased to inform you that your manuscript has been deemed suitable for publication in PLOS ONE. Congratulations! Your manuscript is now with our production department. 

Kind regards, 

on behalf of

Dr. Shamsuddin Shahid 

Academic Editor

PLOS ONE